# Quantifying Similarity of Dynamic Brain Networks: Two Novel Indices for Structural Change and Temporal Evolution

**DOI:** 10.3390/bioengineering12111218

**Published:** 2025-11-07

**Authors:** Xiaocheng Wang, Yongquan He, Tian Zhou, Li Zhang, Shan Fang, Runjie Ni, Weidong Chen, Ruidong Cheng, Xiangming Ye, Dongrong Xu

**Affiliations:** 1College of Computer Science and Technology, Zhejiang University, Hangzhou 310027, China; xiaochengnew@zju.edu.cn (X.W.); 12221046@zju.edu.cn (T.Z.); 2School of Psychology and Cognitive Science, East China Normal University, Shanghai 200062, China; 51184700041@stu.ecnu.edu.cn; 3Department of Rehabilitation Medicine, Zhejiang Provincial People’s Hospital, People’s Hospital of Hangzhou Medical College, Hangzhou 310014, China; zhangli0407@zju.edu.cn (L.Z.); fsphiaa@zju.edu.cn (S.F.); chengruidong@hmc.edu.cn (R.C.); 4Hangzhou Shenyulan Medical Tech Co., Ltd., Hangzhou 310018, China; angelni2010@gmail.com; 5Qiushi Academy for Advanced Studies, Zhejiang University, Hangzhou 310027, China; chenwd@cs.zju.edu.cn; 6Department of Psychiatry, Columbia University & New York State Psychiatric Institute, New York, NY 10032, USA

**Keywords:** dynamic network, index, time points, temporal evolution, structural changes, brain

## Abstract

Brain functional connectivity evolves dynamically during brain development, aging, illness, and cognitive activities. Traditional methods rely on static network snapshots, which do not capture the dynamics of the brain. We propose two new indices: Dynamic Network Similarity (DNS) to measure both temporal and structural dynamic similarity and Dynamic Network Evolution Similarity (DNES) to specifically measure the temporal evolution of dynamic networks. Performance was tested using simulated dynamic networks controlled by four variables (Δφ, λ, α, and β) concerning evolution variations in phase, relative amplitude, noise power, and the span of connectivity strength, respectively. Furthermore, real-world fMRI data from 25 stroke patients pre/post transcranial direct current stimulation (tDCS) rehabilitation were used to test the indices. Patients were randomly sub-grouped into tDCS1 and tDCS2. DNS and DNES thus compared those who received the same therapy (ST: tDCS1 versus tDCS2) and those who received different therapies (DT: tDCS1 versus sham-tDCS). The results showed that DNS was sensitive to all dynamic features, and DNES was primarily sensitive to Δφ and λ. Both indices were able to detect overall difference and capture significantly higher similarity in the ST groups than in the DT groups. Briefly, DNS and DNES appear to be effective tools for studying dynamically evolving brain networks, and may serve as alternatives to traditional static methods. They are particularly useful for analyzing longitudinal neuroimaging data in contexts such as neurodevelopment, aging, and recovery from illness.

## 1. Introduction

Functional magnetic resonance imaging (fMRI), as a non-invasive imaging technology, has been popularly used for investigations in cognitive neuroscience, clinical neurological, and the psychiatric sciences. fMRI detects the neural activities in the brain based on its blood oxygen level-dependent signals. Brain regions integrate and communicate with each other, constituting a number of complex interconnected networks—namely, functional networks—making the brain function as an integrated system when executing cognitive functions and daily life tasks [1]. To characterize the cerebral functional architecture, a brain network can be depicted using a graph theory-based diagram, which comprises of a set of nodes and edges connected the nodes [2,3]. The nodes usually represent the brain regions, and edges represent functional or effective connectivity. Mathematically, the graphical brain networks are frequently formatted as adjacency matrices for further quantitative analysis [4]. Each element in the adjacency matrix represents the relationship between two brain nodes, corresponding to the row and column in the adjacency matrix, respectively. Subsequently, various network indices, for example, degree centrality, network efficiency and small-worldness, can be computed to quantify and study different aspects of the functional networks [5,6,7,8]. Studies following this schema are common in investigations of aging [9], neurological diseases (e.g., Alzheimer’s disease [10,11,12], schizophrenia [13,14], or epilepsy [15]), psychiatric diseases (e.g., autistic spectral disorders [16,17] or major depressive disorder [18]), and brain development [19,20]. Investigators have summarized a number of the most useful and frequently used network indices concerning network efficiency, modularity, and hierarchy [7]. In brief, graph theory-based network analysis has been an indispensable part of studying human brain structure and functions.

People who suffer from neurological diseases such as stroke, schizophrenia, and Alzheimer’s often experience accompanying disruption of brain functional networks. To quantitatively compare the functional networks of different population groups (e.g., healthy people and patients), both global and local feature-based methods have been developed [10,13,21,22,23]. Global feature-based methods assess the overall topological similarity between functional networks by extracting key features (e.g., principal components or network metrics) and comparing them in a reduced feature space [24,25]. Local feature-based methods, in contrast, focus on the correspondence of nodes or edges between networks, using indices such as the Dice coefficient and Jaccard coefficient to quantify overlap [26,27]. While effective, these methods often rely on network binarization, which can discard meaningful information and ignore biologically relevant negative functional connectivity [28,29,30,31].

In the advanced cognitive system, the pattern of brain networks constantly evolves along with network reconfiguration due to constant activities in the brain [32]. For example, one longitudinal study reported decreased intra-network functional connectivity within the default mode network and the executive control network, and the internetwork functional connectivity between these networks then increased and later decreased with aging [33]. Such dynamic evolution of brain networks could not be handled by traditional static network analysis but instead had to be analyzed at separate time points. In contrast, dynamic network analysis would be much more suitable for addressing the challenge of comprehending the physiological patterns that substantiate dynamic brain activities [34,35]. Dynamic functional connectivity analysis has therefore emerged as a method to characterize the variability in the strength or spatial dynamic organization of a dynamic brain network, and is expected to reflect variations in the temporal dynamic features of intersubject brain functional reconfiguration [36]. Hutchison et al. have proposed a method to measure the dynamic processes of brain networks by sliding a time window into the entire fMRI scanning sequence and calculating the functional correlation (FC) in each time window [37]. This method works for scans that last a relatively long time and that have collected a long series of imaging data within one session, but not those that have collected short series of imaging data. Another index, intersubject functional correlation (ISFC), has been proposed for measuring the dynamics of brain networks. It screens intrinsic neural dynamics and separates non-neuronal artifacts, thereby reliably identifying the dynamics of default mode network correlation patterns [38]. These approaches proposed interesting ideas for dynamic network analysis and intersubject comparison. However, the proposed comparisons are semi-quantitative for the functional connectivity or functional network state shifting across brains, meaning that they are unable to provide a completely quantitative result that integrates the dynamic features of dynamic networks across brains. These methods are likely to introduce subjective elements into the results because of their subjective strategy of choosing between dynamic features and functional correlation.

To better characterize the dynamic changes in the evolving patterns of these networks, including their structural changes and temporal evolution, we propose a novel index termed dynamic network similarity (DNS), in which the similarity measurement of the dynamic networks takes into consideration both structural and temporal features. To directly characterize the temporal evolution of the dynamic network concerning its temporal synchronization of the changes but not its structural changes, another index termed dynamic network evolution similarity (DNES) is also developed in this study. Applying these indices to both simulated datasets and the real-world data of stroke patients, we quantified their performance and the results demonstrate that the two new indices appear to be useful alternatives for comprehensively analyzing and understanding the dynamics of evolving brain networks.

## 2. Materials and Methods

Suppose a dynamic network initially consists of a set N of nodes and a set E of edges. The nodes and edges of a dynamic network usually evolve over time, denoted by a time factor t, so that G(N(t), E(t)) defines a dynamic network G at time point t (Figure 1a). The networks in the intermediate states of the dynamic procedure at each time point can be quantified using an adjacency matrix. Each row and column of the adjacency matrix represents a node of the network. The elements in the matrix represent the connectivity strength between the nodes of the network identified by the corresponding row and column (Figure 1b).

Unlike static networks, both structural features and temporal evolutionary features should be considered in the similarity measurement of the dynamic networks. The structural features describe the topological status of the dynamic network, and the temporal evolutionary features describe the temporal status of its connectivity strength. In this study, we further break down these features into four sub-features: (1) The evolutionary trend refers to the consistency of temporal fluctuation trends within brain networks. The phase difference (Δφ) determines the degree of similarity in the evolving patterns between networks. (2) The evolving relative amplitude represents the consistency of temporal fluctuation amplitudes within brain networks. The amplitude ratio (λ) controls the similarity in fluctuation magnitude between the networks over time. (3) Structural topological distribution describes the structural topological organization of brain networks. In the subsequent simulation experiments, noise was added to perturb the topological structure, aiming to examine the effectiveness of DNS and DNES in capturing the structural topological distribution. Noise power (α) refers to the intensity of the Gaussian noise. (4) Connectivity strength span captures the diversity of connection strengths within a brain network, indicating whether the network comprises uniformly strong connections. In the subsequent simulation experiments, we scaled all edge weights in the initial structure of one dynamic network by the factor of the strength span ratio (β).

### 2.1. DNS

DNS takes into consideration both temporal evolution and structural changes. In previous studies, the Pearson correlation coefficient is often used to measure the correlations of changing trends between time series [39,40]. It can also measure the similarities of structural topological distribution between static networks by reorganizing their adjacency matrices into long vectors. We combine these ways of measuring the similarity of evolutionary trends and structural topological distribution between two dynamic networks (Figure 2a). Specifically, we reshape each adjacency matrix of the intermediate network at each time point of the dynamic network into a vector. All vectors of the intermediate networks are linked end-to-end along the time axis to obtain a very long vector *V* of the dynamic network (Figure 2b). Apparently, this long vector *V* contains all of the information of the network. For two dynamic networks *A* and *B*, we define the Pearson correlation coefficient between their long vectors as their similarity of evolutionary trends and structural topological distribution, where a and b are the binary elements in the long vectors *V_A_* and *V_B_*, respectively, supposing the total number of possible connections (edges) involved in each network is *m*.(1)$$Corr\left(V_{A},V_{B}\right)=\frac{\sum_{i=1}^{m}\left(a_{i}-\bar{a}\right)\left(b_{i}-\bar{b}\right)}{\sqrt{\sum_{i=1}^{m}{\left(a_{i}-\bar{a}\right)}^{2}}\sqrt{\sum_{i=1}^{m}{\left(\;b_{i}-\bar{b}\right)}^{2}}}$$

Note that the Pearson correlation coefficient is not sensitive to the amplitude scales even if the connectivity is not binary. In other words, no matter how many times the connectivity strengths of the networks change, the Pearson correlation coefficient of the two networks will not change, as long as it is not zero. For example, the Pearson coefficient of *V_A_* and α**V_B_* remains the same no matter what value α takes, as long as α ≠ 0. Therefore, it is also necessary to introduce a correction term for measuring the similarity of the evolving relative amplitude and connectivity strength span of the dynamic networks. The standard deviation (SD, *σ*) can reflect the dispersion of data and the amplitude of time series fluctuations. The variance of the long vector *V* of one dynamic network can be written as(2)$$\sigma^{2}\left(V\right)=\frac{1}{nm}\sum_{i=1}^{n}\sum_{j=1}^{m}{\left(V_{ij}-\bar{V}\right)}^{2}$$

The *V_ij_* represents the *j*-th (*j* = 1,…, m) edge in the transient network at the *i*-th (*i* = 1,…, n) time point of the dynamic network. The $\bar{V}$ is the mean value of the elements of the long vector. If the numbers of the nodes and edges in the dynamic networks remain unchanged across each time point, Equation (2) can be simplified to(3)$$\sigma^{2}\left(V\right)=\frac{1}{nm}\sum_{i=1}^{n}\sum_{j=1}^{m}{\left(V_{ij}-\bar{V_{i}}\right)}^{2}+\frac{1}{n}\sum_{i=1}^{n}{\left(\bar{V_{i}}-\bar{V}\right)}^{2}$$
where $\bar{V_{i}}$ is the mean value of all edges of the intermediate network at the *i*-th time point, and it can be regarded as the mean of the dynamic network strength at the *i*-th time point. Since the number of edges m of the intermediate network across all the time points is fixed,$\;\bar{V}$ in Equation (3) is equivalent to the mean value of $\bar{V_{i}}$. The variance of the long vector *V* consists of two parts, and the second term is actually the variance of the average edge strengths of the dynamic network across all time points; namely, the evolving relative amplitude. The first term measures the variance of the individual edge strengths involved in each individual intermediate network, which reflects the connectivity strength span of the individual edge strengths in the entire imaging process. Therefore, the *σ* of the long vector *V* combines the measures of the evolving relative amplitude and connectivity strength span of the dynamic network, and the similarity of the evolving relative amplitude and connectivity strength span between two dynamic networks can be measured as the ratio of the *σ* of their long vectors *V_A_* and *V_B_*. After all, the DNS combines the Pearson correlation coefficient term and SD, and is normalized to the range between 0 and 1 as follows:(4)$$DNS=\frac{1}{2}\left(Corr\left(V_{A},V_{B}\right)\times \frac{\min\left(\sigma (V_{A}),\sigma \left(V_{B}\right)\right)}{\max\left(\sigma (V_{A}),\sigma \left(V_{B}\right)\right)}+1\right)$$
where $\sigma \left(V_{A}\right)$ is the SD of all elements of the long vector V. The greater the DNS value, the stronger the similarity between dynamic networks. The normalization may facilitate universal comparisons across different patients, groups, or studies.

### 2.2. DNES

Rhythms of the temporal evolution of dynamic networks are another important aspect worthy of noting when brain networks are being compared. To address this need, we propose another novel similarity index termed DNES. DNES focuses on measuring the evolutionary similarity, or synchronization, of network dynamics.

Static similarity indices usually compare two networks based on the overlap of their corresponding nodes or edges. Dynamic networks are considered to be similar if their evolutionary behaviors are similar, which means the connectivity strengths of their corresponding edges evolve with synchronization towards the same direction. The evolutionary similarity between the corresponding edges is also measured by combining the Pearson correlation coefficient and the SD. In particular, we extract the time series of each corresponding edge evolving with the dynamic network (Figure 2c). The evolutionary similarity between the *i*-th pair of corresponding edges can be defined as(5)$$S_{i}=Corr\left(E_{A_{i}},\;E_{B_{i}}\right)\times \frac{\min\left(SD\left(E_{A_{i}}\right),SD\left(E_{B_{i}}\right)\right)}{\max\left(SD\left(E_{A_{i}}\right),SD\left(E_{B_{i}}\right)\right)}$$

In Equation (5), $E_{A_{i}}$ and $E_{B_{i}}$ are the connectivity strengths, representing the time series of the *i*-th corresponding edges of the dynamic networks *A* and *B*. Subsequently, the evolution similarity of the dynamic networks is integrated by taking the mean value of evolutionary similarity of all corresponding edges as follows:(6)$$DNES=\frac{1}{2}\left(\frac{1}{n}\sum_{i=1}^{n}S_{i}+1\right)$$
where n is the number of edges involved in the dynamic networks. Similarly, DNES is also normalized between 0 and 1, with values closer to 1 indicating higher similarity and those closer to 0 indicating lower similarity. Again, the normalization is prescribed for facilitating universal comparisons.

### 2.3. Computational and Methodological Considerations for DNS and DNES

The computational complexity of both DNS and DNES is $\;O(n^{2}T)$, where *n* is the number of nodes in the network and *T* is the number of time points. This complexity stems from either processing a single long vector of a length proportional to *n*^2^
*T* (for DNS) or iterating through all $O(n^{2})$ edges to perform an O(T) calculation on each edge’s time series (for DNES). The performance bottleneck is the quadratic scaling (*n*^2^) with the number of nodes. Consequently, these methods are computationally feasible and practical for full-brain connectomes based on standard brain atlases with 100–400 nodes. Therefore, DNS and DNES are well-suited to node-based dynamic connectome analysis.

The normalization of DNS and DNES to the 0–1 range is a linear rescaling that enhances interpretability and cross-study comparability. Because the product of the Pearson correlation (−1 to 1) and the standard deviation ratio (0 to 1) naturally lies within [–1, 1], the affine transformation maps this interval to [0, 1], where 1 denotes maximal similarity (perfect correlation with matched amplitudes) and 0 denotes maximal dissimilarity (perfect anticorrelation with matched amplitudes). This normalization provides intuitive thresholds and enables the indices to function as probability-like similarity scores in downstream analyses. The sensitivity of DNS and DNES to normalization choices is generally low, as long as the transformation preserves the original ordering and sign. The adopted linear scaling alters only the range, not the relationships, ensuring interpretability and consistency. In contrast, nonlinear or asymmetric (e.g., logarithmic or rank-based) schemes may distort the distribution by overemphasizing specific similarity ranges. Therefore, linear normalization represents a robust and balanced approach for consistent cross-study applications.

### 2.4. Traditional Similarity

Traditional indices are mostly for comparing network snapshots at a particular moment without incorporating the characteristics of temporal evolution. Indeed, no previous peer indices are ready for us to compare in terms of performance. We therefore include several commonly used traditional similarity indices for static networks for indirect comparisons.

#### 2.4.1. Dice Coefficient

The Dice coefficient measures the structural overlap between two networks, as follows:(7)$$DSC=\frac{2\left|E_{A}\cap E_{B}\right|}{\left|E_{A}\right|+\left|E_{B}\right|}$$
where *E* is the set of edges of the network, and the operator | · | is the number of elements in set *E*. The values of *DSC* range from 0 to 1, with 0 denoting no overlap and 1 complete overlap.

#### 2.4.2. Jaccard Coefficient

Similarly to the notations for the Dice coefficient, the Jaccard coefficient is defined as(8)$$JCD=\frac{\left|E_{A}\cap E_{B}\right|}{\left|E_{A}\cup E_{B}\right|}$$

*JCD* values fall within [0, 1], with 0 for completely different and 1 for identical.

#### 2.4.3. Spectrum Similarity

Spectral distance calculates the difference between two networks based on the feature spectrum of static networks.(9)$$SD=\sqrt{\sum_{i=j=1}^{n}{\left(\lambda_{i}^{A}-\lambda_{j}^{B}\right)}^{2}}$$
where λ*_i_^A^* and λ*_j_^B^* are the eigenvalues of the Laplacian matrix, and $L=D^{-1/2}(D-M)D^{-1/2}$ of the adjacency matrix. The *M* in the equation of the Laplacian matrix is the adjacency matrix of the network snapshot, and *D* is a diagonal matrix whose diagonal elements are the degrees of each node in the network, i.e., the numbers of edges each node connects to within the network. For comparison purposes, the spectral distance can be normalized to(10)$${SD}_{norm}=\sqrt{\frac{\sum_{i=1}^{n}{\left(\lambda_{i}^{A}-\lambda_{i}^{B}\right)}^{2}}{max\left(\sum_{i=1}^{n}{\left(\lambda_{i}^{A}\right)}^{2},\sum_{i=1}^{n}{\left(\lambda_{i}^{B}\right)}^{2}\right)}}$$

When *A* and *B* are the same, ${SD}_{norm}$ = 0; when either $\lambda_{i}^{A}$ or $\lambda_{i}^{B}$ is zero while the other is non-zero, it reaches its maximum value.

Finally, the spectrum similarity is defined as(11)$$SS=1-{SD}_{norm}$$

Therefore, SS values ranges from 0 to 1, where 0 means not similar and 1 means completely the same.

#### 2.4.4. Pearson Correlation Coefficient

The Pearson correlation coefficient of two networks is performed based on their vector forms (see Equation (1)), and its normalized form is(12)$$Corr\left(\vec{v_{1}},\vec{v_{2}}\right)=\frac{Cov(\vec{v_{1}},\vec{v_{2}})}{\sigma_{\vec{v_{1}}}\sigma_{\vec{v_{2}}}}$$
where $Cov(\vec{v_{1}},\vec{v_{2}})$ is the covariance between the two vectors of the networks, and σ is the SD of the vectors. And *Corr* can be normalized to(13)$${Corr}_{norm}=\frac{1}{2}\left(1+Corr\right)$$

The value of ${Corr}_{norm}$ ranges from 0 to 1, with 0 meaning no similarity, while 1 means fully correlated or identical.

### 2.5. Simulation Experiments

In this study, two sets of simulation experiments were designed to check the performance of DNS and DNES. The first set of experiments aimed at testing their sensitivity to both structural changes and temporal evolution (evolutionary trend, evolving relative amplitude, structural topological distribution, and connectivity strength span) in the dynamic networks. The other set of experiments intended to explore whether DNS and DNES were sensitive to the more general (or random) changes in dynamic networks.

Undirected and weighted networks were employed as the network model in the simulation experiments, which mimicked the settings in most brain network studies. The experiments were designed based on the controlled variable method. We first generated two undirected and weighted dynamic networks 1 and 2, which shared the same structural changes and temporal evolution: (1) The two networks at the initial time point were randomly generated but with the same settings in terms of the initial number of nodes and edge values (connectivity strengths). (2) The two networks employed the same evolutionary functions for simulating the evolution of the dynamic networks over time. The edge value F(*t*) of the dynamic network at each time point was decided by(14)$$F\left(t\right)=F\left(0\right)+g\left(t\right)+\varepsilon$$
where F(0) was the edge value at the initial time point 0 of the dynamic process, g(t) is the evolutionary function of the dynamic network, and ε is a noise term. The evolutionary function g(t) was designed based on the Sine function (Equations (15) and (16)), with the consideration that the Sine wave may allow for a clear controllable observation of the impact of phase, amplitude, and translational differences on network connectivity. Therefore, two identical dynamic networks could be generated when the settings of the initial networks and evolutionary functions were the same. In the experiments, both the numbers of nodes and time points of the two dynamic networks 1 and 2 were initially set to 10.

To test the performance of the DNS and DNES in terms of measuring the similarity based on each feature of the dynamic networks, we changed the features of the dynamic networks by controlling their evolutionary functions. The evolutionary functions of the *i*-th edge at time t in dynamic networks 1 and 2, respectively, were as follows:(15)$$g_{1i}\left(t\right)=a_{1i}\sin\left(t+\varphi_{1i}\right)$$(16)$$g_{2i}\left(t\right)=a_{2i}\sin\left(t+\varphi_{2i}\right)$$
where *a* and *φ* are the amplitude and phase of the evolutionary functions, respectively.

To test the robustness of DNS and DNES, three different levels of Gaussian noise with powers of 0.1, 0.01, and 0.001 were applied, respectively. The corresponding signal-to-noise ratio (SNR) was(17)$$\mathrm{SNR}={10\times \log}_{10}\frac{P_{S}}{P_{N}}={10\times \log}_{10}lSNR$$
where $P_{S}$ and $P_{N}$ are the powers of the signal and noise, respectively. In our experiments, the simulated dynamic networks were generated and constrained with the power of their edges being around 1. Therefore, the SNR of the simulation data were 10 dB, 20 dB, and 30 dB when the Gaussian noise was introduced with powers of 0.1, 0.01, and 0.001, respectively. It should be noted that noise in fMRI data follows a Rician distribution rather than a Gaussian one. However, it has been shown that the noise becomes approximately Gaussian when the linear SNR (lSNR) is greater than 2.0 [41], which is equivalent to about 3.0 dB.

#### Experimental Design

In the simulation experiments, four variables (Δφ, λ, α, and β), representing phase evolution, relative amplitude, noise power, and connectivity strength span, were selected to balance physiological plausibility, alignment with prior studies, and empirical validation through pilot simulations. Δφ varied from 0 to π to capture the full spectrum from complete in-phase to anti-phase evolution, simulating realistic phase shifts in oscillatory coupling observed in neural systems. λ ranged from 0 to 1 to model relative amplitude scaling, from weak to fully matched responses, paralleling amplitude modulations in BOLD dynamics and variability in neural connectivity strength. For structural perturbations, α defined the power of zero-mean Gaussian noise added to the initial topology, spanning from low to high levels to mimic realistic SNR conditions and to test robustness. β scaled the overall connectivity strength, with values covering small-to-large multiplicative increases, analogous to the modulation of coupling strength across brain regions as reported in intervention or task-based studies [42]. These parameter ranges were validated in pilot simulations to ensure both the sensitivity of DNS/DNES and stable statistical estimation across 200 repetitions [40].

(A) Experiments for feature sensitivity

Amplitude *a* in Equations 15 and 16 reflects the evolving relative amplitude of dynamic networks, and phase *φ* determines the evolutionary trend of the dynamic networks. Therefore, the similarity of the evolving relative amplitude and the evolutionary trend between the dynamic networks can be, respectively, controlled by changing *λ* = *a*_1*i*_/*a*_2*i*_ and Δ*φ* = *φ*_1*i*_ − *φ*_2*i*_. In addition, the similarity of the structural topological distribution and connectivity strength span of the dynamic networks is under the control of the difference in the structural settings between the initial networks. The below experiments were designed to explore the sensitivity of DNS and DNES to each feature of the dynamic networks.

**Experiment 1:** This experiment was designed to test sensitivity to the evolving relative amplitude of the dynamic networks. We kept the initial structure unchanged, with λ = 1 for both dynamic networks, while varying Δφ from 0 to π. This variation led to decreased consistency in the functional evolutionary trend as Δφ increased.**Experiment 2:** This experiment was designed to test sensitivity to evolutionary trends in dynamic networks. Specifically, we maintained identical initial structures with Δφ = 0 in both networks, while varying λ from 0 to 1 between the evolutionary functions. This variation resulted in a decreased function similarity of evolving relative amplitudes as λ decreased.**Experiment 3:** This experiment examined sensitivity to structural topology distributions in dynamic networks. To manipulate network topologies, we varied the initial structural patterns while maintaining identical evolutionary functions across both networks. A zero-mean Gaussian noise with power α was superimposed to perturb one network’s initial topology, ensuring any observed effects were not attributable to differences in connectivity strength span.**Experiment 4:** This experiment was designed to test sensitivity to the connectivity strength span of the dynamic networks. To change the connectivity strength span of the dynamic networks, we enlarged the edge strength of the initial network in one of the dynamic networks to β times and kept using the same evolutionary functions for both the dynamic networks.

To obtain a sufficient size of samples for adequate statistical power, these experiments were all repeated 200 times. Subsequently, the Pearson correlation between DNS/DNES and the aforementioned variables (Δ*φ*, *λ*, α, and β) of each feature was calculated to test the sensitivity of DNS and DNES to these individual variables. In addition, the residual sum of squares (RSS) of the linear regression of the DNS and DNES at each SNR level was calculated to test the robustness of DNS and DNES.

(B) General experiments

A set of extra experiments was also designed to test the general performance of DNS and DNES when random differences in all these variables were involved. There are two known methods to randomize data: (1) introducing random noise (such as Gaussian noise) into the data, taking the power of the noise as a randomization variable; (2) randomizing the temporal order of the data (such as the time sequence), where the randomization depends on the ratio *γ* of the number of disrupted timepoints to the total time points.

**Experiment 5:** Gaussian noise was introduced to randomize Dynamic Network 2 to examine the performance of DNS and DNES. Obviously, the similarity between dynamic networks would decrease because of the noise disturbance. To mimic the case of a complete random dynamic network, we used a power of Gaussian noise large enough to affect Dynamic Network 2. Therefore, we set the power of the Gaussian noise $P_{N}=0.1,\;0.01,\;0.001$, while assuming the power of the signal to be $P_{S}=1$ (see Equation (17)) in this experiment. Therefore, 3 levels of Gaussian noise N (0, σ) with σ = 0.1, 0.01, 0.001 were applied to the data to test the robustness of the DNS and DNES.**Experiment 6:** Randomize the timing sequence of Dynamic Network 2, based on different ratio γ of the number of the disrupted timepoints to the total time points, and examine the performance of DNS and DNES. Both Experiments 5 and 6 were repeated a thousand times each to ensure adequate statistical power. The correlations between σ (In Experiment 5) and ratio γ (in Experiment 6) with DNS and DNES, respectively, were analyzed to test their sensitivity to the random differences.

### 2.6. Real-World Experiments

#### 2.6.1. Participants

Datasets were retrieved from our past studies on patients with stroke who underwent rehabilitation therapies. Twenty-five participants with stroke were enrolled in this double-blind, randomized controlled study. A computer-generated randomization sequence was created by an independent researcher not involved in patient recruitment or assessment to assign participants to either the transcranial direct current stimulation (tDCS) group or the sham-tDCS group. To ensure effective participant blinding, the sham-tDCS protocol delivered an identical setup, including a brief ramp-up and ramp-down of current at the beginning of the session to mimic the initial scalp sensations of active tDCS, but without delivering any further stimulation.

The recruitment criteria were as follows: (1) first-onset stroke, confirmed as cerebral hemorrhage and cerebral infarction by MRI or CT; (2) aged 18 to 70 years; (3) the disease course was 2 weeks to 3 months; (4) hemiplegia with the Modified Ashworth Scale (MAS) of the affected side was ≤Grade I; and (5) the Brunnstrom stage of the affected side was ≥Stage II. Both groups were scanned with resting-state functional magnetic resonance imaging (rs-fMRI) before and after the treatments. All participants agreed to and signed the informed consent form.

#### 2.6.2. Data Acquisition

All the rs-fMRI data were collected on the GE 3.0 T MRI scanner at a local hospital using an echo-planar imaging sequence as follows: 30 transverse slices, repetition time = 2000 ms, echo time = 30 ms, slice thickness = 4 mm, slice spacing = 0.8 mm, flip angle = 90°, a 64 × 64 acquisition matrix, and FOV = 220 mm × 220 mm. T1-weighted structural images were acquired using a rapid gradient echo sequence as follows: repetition time = 5.7 ms, echo time = 1.9 ms, slice thickness = 1 mm, flip angle = 12°, a 256 × 256 acquisition matrix, and FOV = 240 mm × 240 mm. Participants were asked to lie flat in the imaging instrument, close their eyes, without thinking about any specific things, but keep from falling asleep, and stay still throughout the scan procedure.

#### 2.6.3. Data Processing

To reduce the structural inconsistency of the lesions and increase the statistical power for testing our proposed methods, we flipped the brain images so that the lesions were assigned to the same (left) side of the brains. The Data Processing Assistant for Resting-state fMRI toolbox was used in processing the data (http://www.restfmri.net/forum/DPARSF, 30 August 2025) with the following steps. The first ten volumes of each fMRI dataset were discarded to ensure steady-state longitudinal magnetization. The remaining volumes were slice-timing-corrected and head motion-corrected. Individual datasets were excluded if excessive head motion (rotation > 3 degrees or translation > 3 mm) were detected in the imaging data either before or after treatment. The functional images were normalized to the Montreal Neurological Institute (MNI) standard space by an EPI template and smoothed by a Gaussian smoothing kernel with a 6 mm half-height width. Then, linear drift was removed. Subsequently, head motion was estimated using the Friston 24-parameter model, the noise of the white matter signal and the noise of the cerebrospinal fluid signal were used as covariates in regression analysis to remove nuisance effects. Finally, to reduce low-frequency drift and high-frequency noise, a band-pass filter of 0.01 to 0.08 Hz was applied to all data.

To study the rehabilitation of the motor network of stoke patients under different treatment methods, a total of 20 motion-related cortical regions of interest (ROIs) were created with 6 mm radius spheres around the predefined MNI coordinates. The Pearson correlation coefficients between the fMRI signals of all these ROIs were calculated as their functional connectivity, and confirmed a 20 × 20 symmetric correlation matrices for each patient. The resulting Pearson correlation coefficients were transformed into z-scores by Fisher-Z transformation to satisfy the requirement of normal distribution for further statistical analysis. Finally, the correlation matrices were binarized for traditional similarity indices. The threshold of binarization adopted in this study was set as the statistical parameters a *p*-value of the significance of the Pearson correlation equal to 0.05 (after Bonferroni correction).

#### 2.6.4. Statistical Analysis

DNS and DNES scores were first computed within subjects to characterize the similarity of dynamic motor networks between the pre-treatment and post-treatment sessions for each individual. This within-subject computation captures the extent of rehabilitation-related reorganization in a given patient’s motor network. To validate the effectiveness of the proposed indices in reflecting treatment-specific rehabilitation patterns, we further conducted between-subject comparisons. A reasonable hypothesis was that the rehabilitation of the motor networks of the patients treated with the same therapy (ST) should be more similar than that would be seen in those who were treated with different therapies (DTs). Based on this prior hypothesis, we randomly divided the tDCS group into two subgroups, the tDCS1 group (n_1_ subjects) and the tDCS2 group (n_2_ subjects). Thus, the patients between the tDCS1 group and the tDCS2 group were regarded as ST, and the patients between the tDCS1 group and the sham-tDCS group (n3 subjects) were taken as DT. To test the performance of the DNS and DNES regarding their capability to measure the similarity of dynamic networks, we conducted the following group analyses: (1) The DNS and DNES of the dynamic motor network between each pair of participants in the ST analysis (n1× n2 pairs) were obtained to verify whether the proposed indices may reflect the high similarity in dynamic motor networks as a result of receiving the same treatment. (2) The DNS and DNES of the dynamic motor network between each pair of subjects in the DT analysis (n1× n3 pairs) were compared to examine whether the indices would show lower similarity between the dynamic motor networks as a result of employing different treatments. (3) Whether there were significant differences in DNS and DNES between the ST and the DT analyses, which would implicate a different rehabilitation process of the motor network as a result of receiving either the same or different treatments.

A paired *t*-test was performed to identify the motor network that changed significantly in the tDCS group after the treatment compared to before the treatment. A more lenient statistical threshold (*p* < 0.05) was employed to serve the purpose of effectively comparing the networks by preventing the motor network scale from being excessively limited. A one-sample Wilcoxon signed rank test was performed on the DNS and DNES of the dynamic motor network for the ST patients and the DT patients, respectively, to examine the intra-group similarity. A Mann–Whitney U test was performed on the DNS and DNES of dynamic motor network between the ST patients and the DT patients, respectively, to determine whether there was a difference in similarities between the ST and DT patients (Figure 3). To compare with the performances of the traditional similarity indices, which can only be used for network snapshots at individual timepoints, the means of the traditional similarity indices at all timepoints of the dynamic networks were defined as the similarity between dynamic networks in our study. To assess the statistical effect of the DNS and DNES indices, Cohen’s effect size was also calculated. The d value is usually thresholded at three levels, d = 0.2, d = 0.5, and d = 0.8, respectively, for a small, medium, or large effect.

## 3. Results

### 3.1. Simulation Experiment

#### 3.1.1. Experiment for Feature Sensitivity

For Gaussian noise at all different levels of power, DNS was significantly and negatively correlated with Δ*φ* between the dynamic networks (Figure 4a, N(0, 0.1): r = −0.937, *p* < 0.001, RSS = 0.569; N(0, 0.01): r = −0.969, *p* < 0.001, RSS = 0.400; and N(0, 0.001): r = −0.972, *p* < 0.001, RSS = 0.380); significantly and positively correlated with λ (Figure 4b, N(0, 0.1): r = 0.944, *p* < 0.001, RSS = 0.272; N(0, 0.01): r = 0.982, *p* < 0.001, RSS = 0.115; and N(0, 0.001): r = 0.986, *p* < 0.001, RSS = 0.100); significantly and negatively correlated with α (Figure 4c, N(0, 0.1): r = −0.900, *p* < 0.001, RSS = 1.014; N(0, 0.01): r = −0.909, *p* < 0.001, RSS = 1.328; and N(0, 0.001): r = −0.913, *p* < 0.001, RSS = 1.264); and significantly and positively correlated with the enlarged factor β (Figure 4d, N(0, 0.1): r = 0.981, *p* < 0.001, RSS = 0.196; N(0, 0.01): r = 0.992, *p* < 0.001, RSS = 0.108; and N(0, 0.001): r = 0.993, *p* < 0.001, RSS = 0.103). All reported results were evaluated at the 95% confidence interval. It is apparent that the correlations between DNS and each variable were susceptible to the power of Gaussian noise, which simulated noise that could be introduced in reality during the construction of the network. In addition, the RSS was not sensitive to the changes in α.

Similarity, DNES was significantly and negatively correlated with Δ*φ* between the dynamic networks at different levels of power of the Gaussian noise (Figure 4a, N(0, 0.1): r = −0.991, *p* < 0.001, RSS = 0.508; N(0, 0.01): r = −0.993, *p* < 0.001, RSS = 1.334; and N(0, 0.001): r = −0.993, *p* < 0.001, RSS = 1.663); significantly and positively correlated with λ (Figure 4b, N(0, 0.1): r = 0.985, *p* < 0.001, RSS = 0.153; N(0, 0.01): r = 0.998, *p* < 0.001, RSS = 0.072; and N (0, 0.001): r = 1.000, *p* < 0.001, RSS = 0.006); but it was not correlated with α and β (Figure 4c,d). All reported results were evaluated at the 95% confidence interval. It can be seen that only the correlations between DNES and the evolution variables Δφ and λ increased as the noise intensity decreased. The RSS values were decreased as α decreased.

#### 3.1.2. General Experiment

Experiment 5 showed that both DNS and DNES decreased as the amplitude σ of Gaussian noise increased, but the correlations were not linear. And, when σ approached 10, both DNS and DNES were close to 0.5 (Figure 5a). In Experiment 6, both DNS and DNES decreased with the increase in γ. Furthermore, DNS values were always higher than DNES values. When Dynamic Network 2 was completely disrupted (γ = 1), DNES was close to 0.5 (Figure 5b).

### 3.2. Real-World Experiment

In the preprocessing of real-world data, one patient lacking post-treatment data was excluded. In addition, excessive motion was detected in two participants in the tDCS group and four in the sham-tDCS group. Therefore, a total of 7 subjects were excluded from the analysis. Finally, the tDCS group contained 11 patients (6 in tDCS1 + 5 in tDCS2) and the sham-tDCS group had 7. Thus, ST comparisons formed 30 pairs and DT 42 pairs.

The following functional connectivities were significantly reduced after tDCS treatment: the connectivity between the right middle frontal cortex (rMFC) and the right supplementary motor area (rSMA) (t = −4.794, *p* < 0.001), between rMFC and the right ventrolateral premotor (rPMv) (t = −4.469, *p* = 0.001), between rMFC and the right primary motor cortex (rM1) (t = −3.045, *p* = 0.012), between rMFC and the left superior cerebellum (lSCb) (t = −2.398, *p* = 0.037), between the left dentate nucleus (lDN) and the SMA_R (t = −2.535, *p* = 0.030), between lDN and the rPMv (t = −4.284, *p* = 0.002), between lDN and the lSCb (t = −3.241, *p* = 0.009), the functional connection between the lM1 and the right superior parietal lobule (rSPL)(t = −2.346, *p* = 0.041), and between the left anterior inferior cerebellum (lAICb) and the right anterior inferior cerebellum (rAICb) (t = −4.084, *p* = 0.002). No significant changes were found in the functional connectivity before and after treatment in the sham-tDCS group. The network composed of these functional connectivities was considered the motor network. Further, the motor network before and after treatment formed the dynamic motor network on which we focus.

Both the DNS and DNES of the rehabilitation process of the motor network between the ST groups exhibited high similarity (DNS: z = 4.679, *p* < 0.001, d = 1.771; DNES: z = 4.494, *p* < 0.001, d = 1.378); the DNS of the dynamic motor network between the DT patients was significantly greater than 0.5 (z = 4.963, *p* < 0.001, d = 1.078), but the DNES was not significantly different from 0.5 (z = −1.282, *p* = 0.200, d = 0.199). In addition, the DNS and DNES of the dynamic motor network between the ST groups were significantly greater than those between the DT groups (DNS: z = 2.941, *p* = 0.003, d = 0.785; DNES: z = 4.198, *p* < 0.001, d = 1.203).

For traditional static network similarity indices, the Dice coefficient and Pearson correlation coefficient of the motor network’s rehabilitation process between the ST groups were significantly greater than 0.5 (Dice coefficient: z = 4.060, *p* < 0.001, d = 0.968; Corr: z = 4.576, *p* < 0.001, d = 1.665). For the rehabilitation process of the motor network between DT groups, the Dice coefficient (z = 2.157, *p* = 0.031, d = 0.341), the Jaccard coefficient (z = −3.495, *p* < 0.001, d = 0.586), and the Corr (z = 5.645, *p* < 0.001, d = 2.053) were significantly greater than 0.5, while SS was not significantly different from 0.5 (z = 1.432, *p* = 0.152, d = 0.254). The difference between the traditional similarity indices of dynamic motor network between the ST groups and the DT groups did not reach a significant level. All reported results above were evaluated at the 95% confidence interval.

## 4. Discussion

In this study, we have introduced two novel indices, DNS and DNES, as a comprehensive set of tools for quantifying and comparing the dynamic similarity of evolving brain networks. DNS simultaneously incorporates both structural changes and temporal evolution by compressing the dynamic network into a long-vector representation and quantifying similarity in evolving amplitude and connectivity strength span. DNES isolates the temporal synchronization of dynamic changes by focusing on the time series of corresponding edges, thereby emphasizing evolutionary trajectories rather than structural configurations. Both simulated and real-world data from the small cohort of stroke patients confirmed the effective performance of the two indices, which outperform commonly used traditional indices for assessing network similarity. Thus, our proposed methods may provide a quantitative basis for future studies on the dynamic similarity of evolving brain networks. However, this is a proof-of-concept study based on a small sample, although the DNS and DNES metrics appear to be promising tools in this small and specific sample. Their general effectiveness will certainly require validation in larger, independent cohorts and against other dynamic methods.

Compared with existing dynamic similarity frameworks, the proposed DNS and DNES provide several advantages. Previous sliding-window and state-based approaches, as formalized in the dynamic functional network connectivity (dFNC) framework, cluster time-varying connectivity into discrete states and compare dwell times or transition patterns, which capture recurring motifs but ignore fine-grained temporal synchronization of the connectivity between nodes in the networks [36,43,44,45]. Clinical applications of this framework, such as the work by Fiorenzato et al. in Parkinson’s disease, have demonstrated its usefulness in identifying disease-related alterations in dynamic connectivity patterns, but these state-based metrics still focus primarily on transitions among predefined states rather than similarity in an evolving procedure at multiple time points in longitudinal studies [46]. Hidden Markov Models (HMMs) further model latent switching processes, but strongly depend on assumptions about the number and form of hidden states, which could unnecessarily introduce biases [47]. Other graph-based measures, such as dynamic modularity and flexibility metrics, highlight reconfigurations of community structure but do not directly quantify network similarity across multiple time points within an individual [48,49]. In contrast, DNS explicitly integrates both the temporal evolution and the spatial (structural topological distribution of networks) information into a single interpretable score, while DNES focuses on temporal synchronization of connectivity time series between all the networking nodes, thus capturing amplitude difference and dynamic fluctuations in a way that state-based or modularity-based indices are unable to offer. Importantly, both DNS and DNES are normalized to the [0, 1] range, facilitating intuitive comparisons and interpretations across different studies in a way that is as easy as understanding probability. These properties make DNS and DNES complementary to existing frameworks, offering a direct and unified measure for assessing dynamic network similarity that is sensitive to both evolving topologies and relative strength variations.

In Experiments 1 to 4, based on simulated datasets, we examined the performance of DNS and DNES in measuring the similarity of the four evolving features (evolutionary trend, evolving relative amplitude, structural topological distribution, and connectivity strength span) of dynamic networks. DNS was able to closely reflect the variations in the dynamic networks due to the four evolving features, which indicated that DNS is sensitive to such temporal evolutionary and structural features. DNES was found to focus on the variations in the dynamic networks in relation to their evolutionary trends and evolving relative amplitudes, indicating that DNES is specifically sensitive to the temporal evolutionary features of the dynamic networks. As we have seen in Experiments 1 and 2, when Δφ and λ approached 0 and 1, the DNS and DNES values were both close to 1, meaning that the dynamic networks evolved in a relatively synchronized fashion. In Experiments 3 and 4, DNS was around 1 when noise was not present in Dynamic Network 2 and the connectivity strength did not change. DNS decreased with increased σ and decreased β, demonstrating that DNS can reflect that the structural changes in the dynamic networks happened in a similar pattern. In these experiments, DNES consistently remained around 1 and showed little sensitivity to the structural changes, suggesting that DNES is only slightly affected by such structural variations in dynamic networks. It is worth noting that the DNS values were higher than the DNES values in evolution-related Experiments 1 and 2. This could be attributed to the fact that DNS contains the measurement of the temporal evolutionary features of dynamic networks whereas DNES neglects them. In addition, under different levels of noise, the RSS values of DNS were not affected significantly but were generally higher than those of DNES. Such a difference may indicate that DNS is more robust; however, with a lower accuracy, compared with DNES.

Note that the insensitivity of DNES to structural variables is not a flaw, but actually an intentionally and successfully designed feature aiming at isolating temporal evolutionary features from others, which is an extension of our previous preliminary work based on DNS [40]. As evident in the experiments, when the network structure was manipulated via noise and connectivity strength, DNS values changed accordingly, whereas DNES values remained stable, indicating that DNES is relatively insensitive to these purely structural perturbations. Yet, later in the experiment, we saw that DNES was indeed able to capture the temporal evolution of the networks. Therefore, this behavior confirmed its successful isolation of temporal features. However, whether this separation is an advantage or an “insensitivity” to relevant dynamics depends on the specific scientific question. On one hand, this specificity makes DNES a powerful analytical tool that is complementary to DNS, which provides a holistic view. It enables researchers to specifically test hypotheses concerning the similarity of a network’s temporal trajectory of change. On the other hand, in many biological systems, temporal evolution is realized through structural changes, and an index that ignores these structural dynamics might miss critical mechanistic information. Thus, while DNES successfully isolates temporal features, its most powerful application lies in its synergistic use with DNS. This allows for a multi-dimensional assessment of dynamic network similarity, capturing both the overall evolutionary process and dissecting its purely temporal patterns.

In the experiments that considered overall changes, both DNS and DNES between Dynamic Network 1 and 2 decreased when Network 2 was randomized by superimposed Gaussian noise disturbance (Experiment 5). This result indicates that DNS and DNES remained working in terms of measuring the similarity between the dynamically evolving networks even when general changes occurred. In Experiment 6, both the DNS and DNES of the two dynamic networks decreased along with the disruption of the time sequence of Dynamic Network 2, and DNS was always higher than DNES. We speculate that the disruption of the dynamic network timing order was the major contributor, as doing so completely disrupted the mild evolutionary trajectory of the network dynamics. The results indicate that, although the aforementioned general experiments did not take into account in their design the fundamental characteristics of dynamic networks, the observed changes in similarity may still be interpreted in terms of the dynamic evolution and structure of the network. This evidence actually indicates that focusing on a few dynamic network features, as proposed by our method, is reasonable and valuable.

For simplicity in demonstrating the principles of DNS and DNES, we used binary networks in this work. In fact, both the DNS and DNES indices are inherently flexible and are ready to be extended beyond binary undirected networks. However, some minor adjustments may be needed. For weighted networks (edges are expressed as connection strengths using continuous real values), the Pearson correlation component in Equations (4) and (6) naturally accommodates edge weights, while the variance-based terms capture the relative amplitude and fluctuation spans of connection strengths, making the indices directly applicable without modification. For directed networks, the adjacency matrix can be revised to incorporate directionality, either using signed values or adding a mechanism to the similarity calculation, enabling DNS and DNES to capture asymmetries in temporal evolution. Moreover, the framework can be generalized to multimodal datasets such as EEG–fMRI, as long as the time series of the data from different modalities can be aligned and thus their adjacency matrices are meaningfully matched with each other and concatenated, which is actually a different issue than the nature of proposing the DNS and DNES indices. We have carried out some work along this direction. Limited by the length of the current paper, we will report this in a separate publication.

The experiments based on real-world data acquired from stroke patients showed that the DNS and DNES of the dynamic motor network in ST groups exhibited high similarity with a substantial effect size (DNS: z = 4.679, *p* < 0.001, d = 1.771; DNES: z = 4.494, *p* < 0.001, d = 1.378). This suggests a high degree of similarity in the motor network rehabilitation processes among the ST groups, in line with the reasonable hypothesis we made at the very beginning. In contrast, using traditional similarity indices, only the Dice coefficient and Pearson correlation coefficient were above 0.5 with statistical significance and a large effect size (Dice coefficient: z = 4.060, *p* < 0.001, d = 0.968; Corr: z = 4.576, *p* < 0.001, d = 1.665), highlighting the outstanding sensitivity of DNS and DNES in detecting the nuances of motor network similarity in ST patients. Testing the DT groups, the DNES of the dynamic motor network did not exhibit a high similarity (DNES: z = −1.282, *p* = 0.200, d = 0.199), inferring that the rehabilitation process of the motor network using different treatment methods could have introduced differences in brain functional network reorganization, thereby leading to weak similarity across the DT groups. This may be attributed to the fact that there was no significant change in the motor network before and after the treatment sessions in the sham-tDCS group, while the motor network in the tDCS group did incur significant changes. As a result, the DNS between the DT groups was not significant. Note that the DNS between the DT groups exhibited high similarity (DNS: z = 4.963, *p* < 0.001, d = 1.078), which may have been caused by a certain degree of structural similarity in the motor networks among the participants in the DT groups.

Biologically, a high DNS value in a context such as that in our case of stroke rehabilitation suggests that tDCS treatment promotes robust neuroplasticity, reflecting convergence in how patients’ motor networks reorganize to support functional recovery. Clinically, DNS and DNES could be incorporated into patient monitoring and follow-up workflows—for example, by using periodic rs-fMRI or EEG to track network reorganization and assess rehabilitation progress or neuromodulation strategies, identifying when a patient’s recovery trajectory deviates from expected patterns derived from prior patients or existing studies. In addition, these indices have framed a way to explore interindividual variability and identify patient subgroups with shared recovery trajectories. By measuring DNS/DNES values that cover the entire session from pre- to post-treatment and the variability of the values across many patients, it becomes possible to either cluster individuals who exhibit similar patterns of network reorganization, revealing potential neural recovery subtypes, or perhaps form an atlas of recovery or evolution trajectory. Individual, interindividual or group-wise variability can be quantified. Such information may help explain differential treatment responsiveness and guide personalized interventions in neurorehabilitation. Moreover, as DNS and DNES are normalized and correlation-based, they can be applied consistently across datasets or even across studies, enabling meta-analytic investigations of recovery patterns in larger patient cohorts. Future research could integrate these indices with clinical or behavioral measures to examine whether patients with higher DNS/DNES share common functional improvements, thereby linking dynamic network patterns to meaningful clinical outcomes.

The results for DNS and DNES imply that neuroplastic reorganization in the DT groups’ motor network rehabilitation predominantly manifests through evolutionary adaptations rather than structural reconfigurations. The empirical results not only validated the measurement validity of DNS and DNES in quantifying rehabilitation process similarity between DT groups, but also revealed distinct operational characteristics between these two metrics in dynamic network assessment. The DNES metric specifically focuses on assessing similarity in evolutionary patterns during motor network rehabilitation across distinct groups, whereas the DNS metric incorporates both evolution and structure into dynamic brain network similarity assessments.

The result of dynamic motor network similarity between the ST groups and between the DT groups showed that the DNS and DNES of the ST groups were markedly higher than those of the DT groups, with a significant effect size, as noted in the preceding paragraph. This outcome aligns with our expectations and underscores the excellent sensitivity of DNS and DNES in capturing the dynamic similarity of evolving networks in the real world. Yet, the traditional static network similarity indices showed no significant differences between the similarity values of the ST groups and the DT groups. The proposed indices may offer a potential way to quantify brain network evolution against a reference, as was outlined earlier. For example, by tracking the dynamic similarity between the brain network of a stroke patient and one calculated based on a group of healthy people, clinicians might be able to obtain complementary information to assist in evaluating the rehabilitation status of the patient and monitor the patient’s functional recovery progress. Beyond their utility in stroke rehabilitation, DNS and DNES potentially have broader applications across diverse domains where dynamic networks play a critical role. For example, studying childhood or adolescent brain development, the disease process of neurological and psychiatric disorders, and brain aging. All these studies involve both structural and temporal reorganization that may typically differ across individuals. This approach thus offers a paradigm that fundamentally differs from traditional network similarity measures, both dynamic and static.

A challenge in clinical practice is that the datasets may frequently see irregular or missing timepoints. Extra preprocessing would be needed in such cases. One solution is applying interpolation techniques to deduce missing values, or by using model-based temporal alignment approaches to achieve correspondence across unevenly sampled time series. These adaptations would preserve the comparability of evolutionary trends and amplitudes across datasets, thereby extending the utility of DNS and DNES in broader network neuroscience contexts. New methods should be explored and developed along this direction.

There are several limitations that should be acknowledged. First, although the equations defining DNS and DNES are mathematically sound on the 0–1 scale, the sensitivity of these measures to different normalization choices has not been systematically evaluated. Future work could assess the robustness of DNS/DNES under alternative normalization strategies. Second, the size of the real-world dataset used in our experiment was relatively small, and six participants were excluded due to data quality issues. This limited sample size reduced the statistical power for between-group comparisons and constrained the generalizability of the findings, although our results demonstrated quite strong statistical significance (*p*-value < 0.001). Non-parametric approaches such as bootstrapping, as well as replication in larger longitudinal cohorts, will be helpful to verify these results and to reinforce their robustness. Third, the present implementation of DNS and DNES relies on Pearson correlation and standard deviation ratios, which assumes linearity and homoscedasticity across time. Datasets in the real world may be more complex; therefore, exploring nonlinear extensions using methods such as Spearman correlation, mutual information, or dynamic time warping, will enhance and broaden the methodological utility of DNS/DNES. Fourth, while the noise in the fMRI data reduced to a Gaussian distribution, the inherited noise on brain networks derived from the data may not follow it perfectly. Our simulation model could have failed to fully capture the temporal autocorrelation and physiological noise. Incorporating more realistic noise components (e.g., respiration- and cardiac-related fluctuations) in future simulations would enhance biological plausibility. Finally, the binarization of networks at a Bonferroni-corrected threshold of *p* < 0.05, although stringent and specific, may eliminate weak yet meaningful connections. Given that DNS and DNES are defined for weighted matrices, applying them directly to continuous connectivity values may preserve additional information and improve sensitivity. This should be systematically pursued in the directional connectivity version as the next step of our work.

## 5. Conclusions

We have proposed two novel indices, termed DNS and DNES, for measuring similarity between dynamic networks, and provided evidence for their performance using both simulated data and real-world data. Compared with traditional similarity methods, DNS and DNES can accurately characterize the structural and temporal evolutionary features of dynamic networks, providing an integrated similarity measurement between different networks that could not previously be captured by traditional measurements. The proposed indices may open new venues for dynamic network analysis and help address the limitations of previous studies that only characterized snapshots of brain networks at static timepoints. We will further verify and refine DNS and DNES, making them more robust and more widely applicable.

## Figures and Tables

**Figure 1 bioengineering-12-01218-f001:**
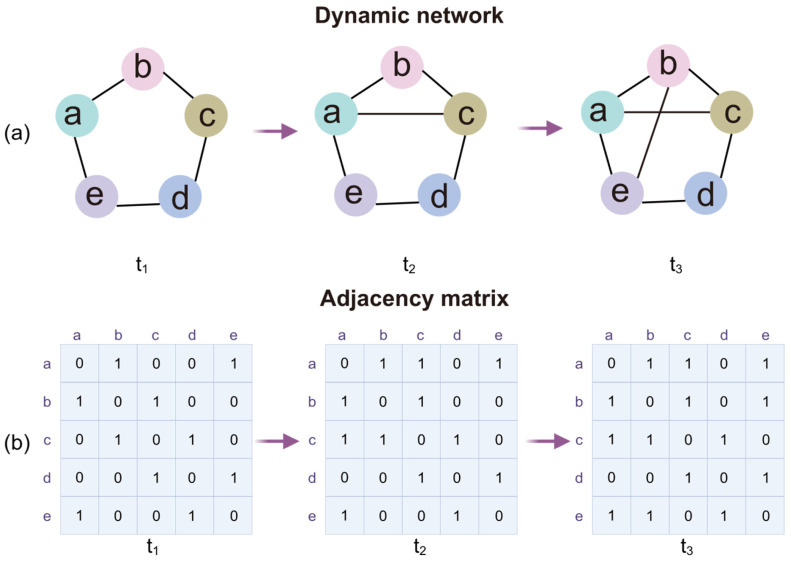
Evolution model of a dynamic network. (**a**) An evolving connectivity process at the network periphery. As an example case, this brain network consists of 5 brain regions, denoted as a, b, c, d and e. (**b**) Adjacency matrices of the dynamic network at the corresponding time points of the evolving process, where the elements 1 and 0 represent connectivity or no connectivity, respectively, between the nodes corresponding to the row and column.

**Figure 2 bioengineering-12-01218-f002:**
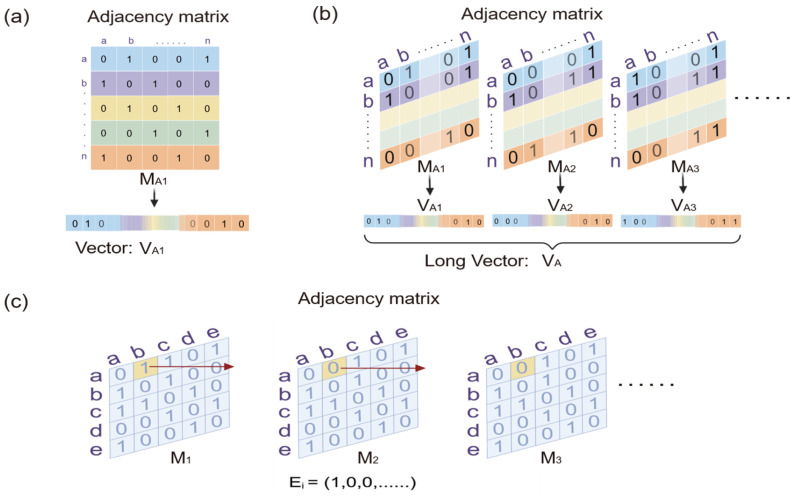
(**a**) The adjacency matrix of the network is reshaped into the form of a vector. Each row or column (from 1 to n) of the adjacency matrix is connected end-to-end. (**b**) The diagram of DNS. The vector of the adjacency matrix at each time point of the dynamic network is connected end-to-end to form a long vector. (**c**) The diagram of DNES. The time series of each element of adjacency matrix is extracted for DNES calculation.

**Figure 3 bioengineering-12-01218-f003:**
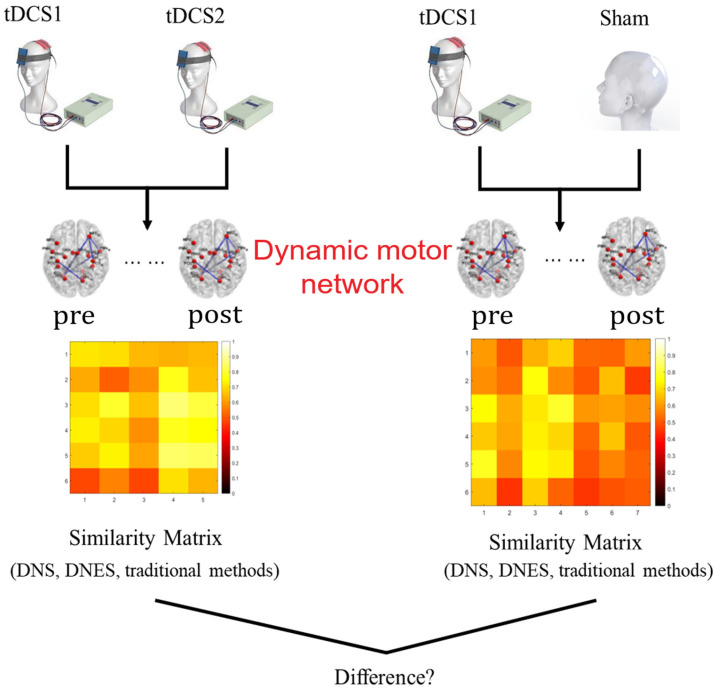
The DNS and DNES of stroke patients. The DNS, DNES, and traditional indices of dynamic motor networks were measured between the ST groups (tDCS1 and tDCS2) and between the DT groups (tDCS1 and sham-tDCS), respectively. The similarity matrix was filled by similarity between individuals intergroup, therefore the sizes of the similarity matrices of the ST groups and DT groups were 6*5 and 6*7, respectively. Statistical methods were employed to determine the similarity levels of the dynamic motor networks of the ST groups and DT groups, and whether the similarities of the ST groups and DT groups were different.

**Figure 4 bioengineering-12-01218-f004:**
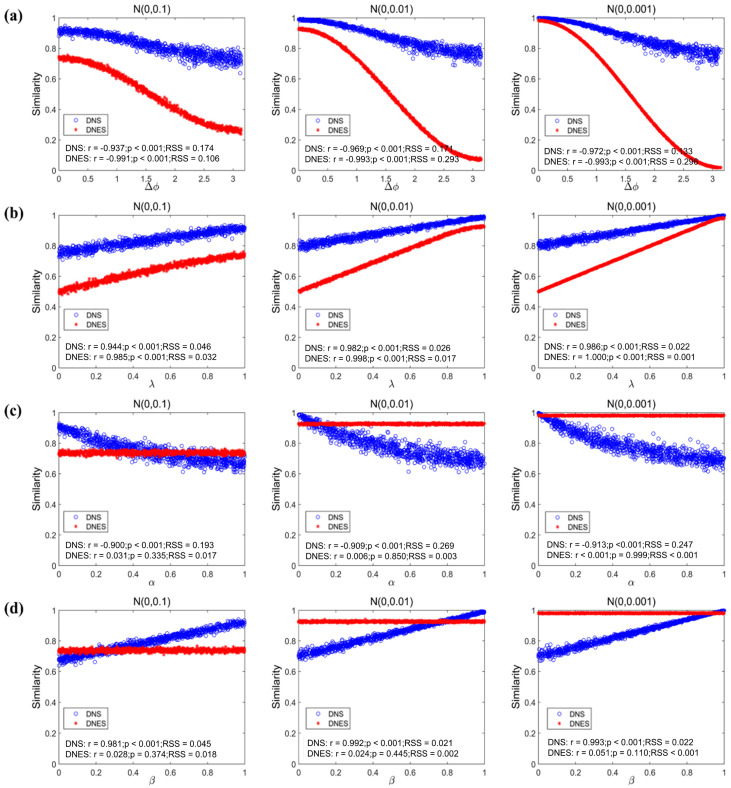
The feature-specific experiments for DNS and DNES. (**a**) Both the DNS and DNES of the simulated dynamic networks decreased with increasing Δφ. (**b**) Both DNS and DNES increased with the parameter λ of the evolutionary function. (**c**) DNS decreased with the increase in α, while DNES appeared not to be affected. (**d**) DNS decreased with the increased of β. The RSS of DNS was not affected by three levels of Gaussian noise (N (0, 0.1), N (0, 0.01), and N (0, 0.001)), whereas DNES exhibited higher sensitivity to α variations. DNS: dynamic network similarity; DNES: dynamic network evolution similarity; RSS: residual sum of squares.

**Figure 5 bioengineering-12-01218-f005:**
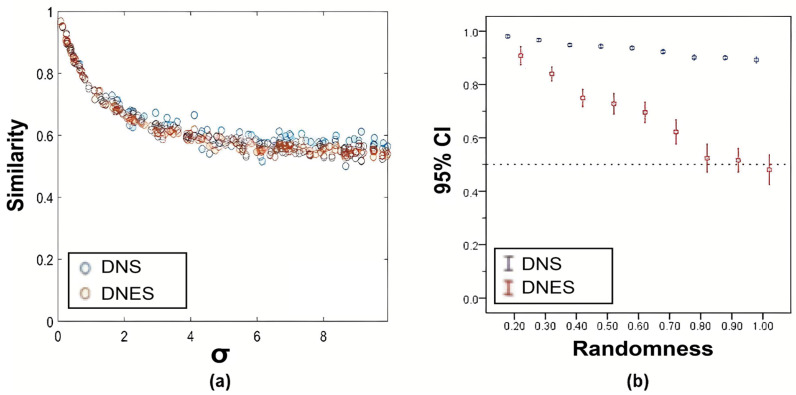
The general experiments for DNS and DNES. (**a**) The DNS and DNES values of the simulated dynamic networks decreased inversely with the increased power σ of Gaussian perturbation (i.e., randomness). (**b**) The DNS and DNES values of the simulated dynamic networks decreased with the increased randomness of its timing sequence. DNS was overall higher than DNES. DNS: dynamic network similarity; DNES: dynamic network evolution similarity.

## Data Availability

The data used to support the findings of this study are included within the article and are available from the corresponding authors upon request. Software and deidentified test datasets related to this paper are available upon request. Please contact the authors for downloading details.

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
