# Peer review of "Quantifying Similarity of Dynamic Brain Networks: Two Novel Indices for Structural Change and Temporal Evolution"

_bioengineering, 2025, doi:10.3390/bioengineering12111218_

Round 1

Reviewer 1 Report

Comments and Suggestions for Authors

This manuscript introduces two innovative indices, Dynamic Network Similarity (DNS) and Dynamic Network Evolution Similarity (DNES) for quantitatively assessing similarity between dynamic brain networks by integrating both temporal and structural characteristics. Using simulation experiments and real-world fMRI data from stroke patients undergoing tDCS rehabilitation, the authors demonstrate that these indices outperform traditional static similarity measures. The topic is timely and relevant, addressing key limitations of static connectivity models in neuroscience. The paper is generally well-organized, methodologically rigorous, and presents a valuable contribution to computational neuroimaging. However, several methodological and conceptual aspects require clarification and refinement before publication. The authors are encouraged to address the following points and revise the manuscript accordingly:

  1. Could the DNS and DNES indices be extended to weighted or directed networks, or to multimodal datasets (e.g., EEG–fMRI)? Please also discuss the computational complexity of these methods for large-scale networks and their feasibility for full-brain connectomes with more than 100 nodes.
  2. The manuscript would benefit from a more explicit comparison with existing dynamic similarity frameworks, such as dynamic graph metrics and time-varying connectivity states.
  3. Clarify the conceptual differences between DNS and DNES—specifically, how the weighting between temporal and structural components is determined and whether the two measures can be combined or cross-validated.
  4. The introduction ends abruptly after line 114. It should conclude with a clear hypothesis or research question to guide the reader.
  5. Discuss potential applications of DNS and DNES beyond stroke rehabilitation to illustrate their broader relevance.
  6. While the equations defining DNS and DNES are mathematically sound, please elaborate on the derivation and rationale for normalizing results to the 0–1 scale. How sensitive are these measures to different normalization choices?
  7. For the real-world data, clarify whether DNS/DNES were computed pre–post treatment within subjects or only between subjects, as this distinction affects the interpretation of “rehabilitation similarity.”
  8. In the simulation section, justify the choice of parameters (Δφ, λ, α, β). Were these based on physiological reasoning, prior studies, or empirical tuning?
  9. The real-world dataset (n = 25) is relatively small. Please discuss its statistical power and implications for generalizability.
  10. Provide more details on the randomization and blinding procedures for the tDCS versus sham groups.
  11. Explain why DNES appears insensitive to structural variables (α, β). Does this reflect a successful isolation of temporal features, or might it indicate insensitivity to relevant structural dynamics?
  12. Expand the discussion of biological meaning—for example, what does a high DNS value imply about neuroplasticity following stroke rehabilitation?
  13. Given the need for temporal correspondence, discuss how the methods could adapt to irregular or missing timepoints (e.g., through interpolation or model-based alignment).
  14. Strengthen the clinical relevance section by providing a practical example of how DNS and DNES could be integrated into patient monitoring or rehabilitation assessment workflows.

Author Response

We would like to sincerely thank all the reviewers for their careful evaluation of our manuscript and their constructive comments. We appreciate their recognition of the novelty and methodological rigor of our work, and we are grateful for the insightful suggestions that help improve the quality of the manuscript highlighting important conceptual, methodological, and interpretative aspects. Please see below our point-to-point feedback for every comment raised by the reviewers.

Comments and Suggestions for Authors

This manuscript introduces two innovative indices, Dynamic Network Similarity (DNS) and Dynamic Network Evolution Similarity (DNES) for quantitatively assessing similarity between dynamic brain networks by integrating both temporal and structural characteristics. Using simulation experiments and real-world fMRI data from stroke patients undergoing tDCS rehabilitation, the authors demonstrate that these indices outperform traditional static similarity measures. The topic is timely and relevant, addressing key limitations of static connectivity models in neuroscience. The paper is generally well-organized, methodologically rigorous, and presents a valuable contribution to computational neuroimaging. However, several methodological and conceptual aspects require clarification and refinement before publication. The authors are encouraged to address the following points and revise the manuscript accordingly:

Answer: Thank you for the comments. Please see below our detailed point-by-point feedback.

  1. Could the DNS and DNES indices be extended to weighted or directed networks, or to multimodal datasets (e.g., EEG–fMRI)? Please also discuss the computational complexity of these methods for large-scale networks and their feasibility for full-brain connectomes with more than 100 nodes.

Answer: Thank you for this valuable suggestion. Both DNS and DNES can be extended to weighted or directed networks by incorporating edge weights or directionality into the similarity computation. However, limited by the length of this paper, these extensions will be reported in another paper. To address the reviewer’s concern, we have added a paragraph discussing the planned extensions of DNS and DNES. (Please see Lines 684- 699, marked as R1C1(for Reviwer#1 Comment #1)). We have also added section 2.3 (Lines 221-228, marked as R1C1, too) to address the computational complexity.

  1. The manuscript would benefit from a more explicit comparison with existing dynamic similarity frameworks, such as dynamic graph metrics and time-varying connectivity states.

Answer: We appreciate this constructive comment. We have added a new paragraph to the Discussion section to provide a more explicit comparison between the proposed DNS/DNES indices and those existing dynamic similarity frameworks. Please see Lines 603-627 in this revision.

  1. Clarify the conceptual differences between DNS and DNES—specifically, how the weighting between temporal and structural components is determined and whether the two measures can be combined or cross-validated.

Answer: Thank you for this insightful comment. We have clarified the conceptual distinction between DNS and DNES in this revision, and also elaborated how DNS and DNES can be jointly analyzed, and the advantages of using them jointly. Please see Lines 590-595 and 659–667 in the updated text.

  1. The introduction ends abruptly after line 114. It should conclude with a clear hypothesis or research question to guide the reader.

Answer: Thank you for pointing out this. We have now added a sentence to the end of the last paragraph of the Introduction section to wrap up the statements. (Lines 105-109)

  1. Discuss potential applications of DNS and DNES beyond stroke rehabilitation to illustrate their broader relevance.

Answer: Thank you for the constructive comment. We have expanded a paragraph in the Discussion to outline potential applications of DNS and DNES beyond studying stroke rehabilitation. (Lines 761-766)

  1. While the equations defining DNS and DNES are mathematically sound, please elaborate on the derivation and rationale for normalizing results to the 0–1 scale. How sensitive are these measures to different normalization choices?

Answer: Thank you for raising this important point. While the values before the 0-1 normalization may distribute in a wide range not directly comparable across different studies or patients, the normalizing of DNS and DNES values actually uniforms the sensitivity of the indices and thus facilitates interpretability and comparison across subjects and datasets. We now have amended the text to explicitly highlight this advantage (Lines 229-242).

  1. For the real-world data, clarify whether DNS/DNES were computed pre–post treatment within subjects or only between subjects, as this distinction affects the interpretation of “rehabilitation similarity.”

Answer: Thank you for your valuable comment. Yes, for the real-world data, the indices were calculated pre-post treatment within subjects before entering statistical analysis to evaluate changes over the course of rehabilitation. We now have clarified this point in the revision. Please see section 2.6.4, Lines 455-460.

  1. In the simulation section, justify the choice of parameters (Δφ, λ, α, β). Were these based on physiological reasoning, prior studies, or empirical tuning?

Answer: Thank you for the comment. The parameters (Δφ, λ, α, β) were chosen based on prior studies to reproduce realistic brain dynamics. Please see the added explanation in section 2.5.1(Lines 328-342).

  1. The real-world dataset (n = 25) is relatively small. Please discuss its statistical power and implications for generalizability.

Answer: Thank you for your comment. We have discussed this issue as a limitation in the Discussion section, noting the relatively small sample size with limited statistical power and generalizability. We also highlight the need of larger follow-up studies to confirm these findings (Lines 781-787).

  1. Provide more details on the randomization and blinding procedures for the tDCS versus sham groups.

Answer: Thank you for the comment. We have added more details about the randomization and blinding procedures to the text. Please see Lines 404-410, and Lines 552-554.

  1. Explain why DNES appears insensitive to structural variables (α, β). Does this reflect a successful isolation of temporal features, or might it indicate insensitivity to relevant structural dynamics?

Answer: Thank you for this insightful comment. We have clarified that the insensitivity of DNES to α and β was a purposed design to isolate temporal dynamics, and suggested possible scenarios that they be used in combination. Please see the fourth paragraph in the Discussion section (Lines 651-669).

  1. Expand the discussion of biological meaning—for example, what does a high DNS value imply about neuroplasticity following stroke rehabilitation?

Answer: Thank you for your comment. We have expanded the Discussion to clarify that in our application case, higher DNS values may indicate more stable and efficient network reorganization, reflecting enhanced neuroplasticity during stroke rehabilitation. Please see Lines 720-722 in the discussion section.

  1. Given the need for temporal correspondence, discuss how the methods could adapt to irregular or missing timepoints (e.g., through interpolation or model-based alignment).

Answer: Thank you for the comment. This is actually a practical case that we need to deal with. Additional preprocessing steps will be needed to handle the missing data points, using data alignment techniques or data interpolation methods. This also suggests a future direction to work on. We have added a paragraph to the Discussion section to address this point. Please see Lines 769-775.

  1. Strengthen the clinical relevance section by providing a practical example of how DNS and DNES could be integrated into patient monitoring or rehabilitation assessment workflows.

Answer: Thank you for this valuable suggestion. We have added practical examples of how DNS and DNES may be integrated into patient monitoring and rehabilitation assessment. Please see Lines 722-727.

Reviewer 2 Report

Comments and Suggestions for Authors

The manuscript proposed two indices, DNS and DNES, for comparing dynamic brain networks. While the intent to move beyond static snapshot analysis is commendable and clinically relevant, the study is undermined by an underrepresented literature review that fails to establish true novelty. There are also some methodological oversights and statistical limitations. 

1. The authors have either missed or chosen not to engage with a body of literature that directly prefigures their work. "DNS" is not a new name. The 2022 paper by He et al. (Sheng Wu Yi Xue Gong Cheng Xue Za Zhi) proposed the exact same acronym "DNS" (Dynamic Network Similarity) for the same purpose (comparing dynamic networks in stroke patients undergoing tDCS). The authors must explicitly cite this paper and clarify how their DNS formulation is mathematically or conceptually distinct. 

2. The field of dynamic network comparison is well established. The authors framed the problem as if no methods exist for comparing dynamic networks. This is incorrect. For example, the review by Mheich et al. (2020) is a dedicated summary of "brain network similarity" methods. The framework by Fu et al. (2018) and the clinical application by Fiorenzato et al. (2019) demonstrated that comparing dynamic states (e.g., dwell time, frequency, transition number) is a standard approach. While DNES is a different implementation, the core idea of quantifying temporal properties for clinical comparison is well-established. The only defensible novelty lies in the specific mathematical formulation of the indices: the particular way they combine Pearson correlation and standard deviation across the entire vector (DNS) or per-edge time series (DNES). The authors must pivot their claim of novelty from the idea of dynamic similarity to the specific performance of their unique algorithmic implementation.

2. The proposed metrics have theoretical ambiguities. DNS conflates temporal and structural information into a single score. The long vector V jumbles together the evolution of each connection over time. A high DNS score could mean two networks have similar structures but are out-of-phase, or have different structures but evolve in a similar way. This makes the score difficult to interpret clinically. What does a DNS of 0.7 actually mean? Is the brain network structurally similar, temporally synchronized, or both? By comparison, state-based analyses (Fiorenzato et al., 2019) are more interpretable: "Patients dwell longer in a segregated state" provides a clear, testable hypothesis about brain function. Similarly, by calculating similarity for each edge and then averaging, DNES is highly sensitive to noise in individual connections. A few noisy or artifact-laden edges can disproportionately drag down the entire similarity score. It assumes that all connections are equally important for measuring "evolutionary similarity," which is likely false. The DNES approach appears to lack the robustness of methods that first identify major network states (like the dFNC framework) and then compare the properties of those robust, aggregate states. The claim that values of 0.5-1.0 represent "high similarity" is a bit arbitrary and not validated on multiple datasets. Conducting additional analyses to show what drives differences in DNS and DNES (e.g., which connections or time periods contribute most to low similarity) would make the indices more useful.

3. The methods section does not justify why the specific mathematical formulation was chosen. Why use a sine wave for simulation? Why is the threshold for "high similarity" set at 0.5? The method for binarizing networks for traditional metrics is mentioned, but the critical choice of threshold (statistical significance after Bonferroni correction) may itself bias the results, as binarization is a known limitation. The real-world fMRI data analysis shows weakness of the study. It started with 25 patients and then excluded 7, leaving only 18. These were then split into tDCS1 (n=6), tDCS2 (n=5), and sham (n=7). Conducting group comparisons with such small, underpowered subgroups is a major statistical flaw. The significant p-values are suspect and likely overfit; the results are unlikely to be replicable. The use of both parametric (t-test) and non-parametric (Mann-Whitney U, Wilcoxon) tests without a clear justification for each choice raises concerns about p-hacking. The primary finding, comparing ST vs. DT groups, uses a Mann-Whitney U test on only 30 vs. 42 data points (similarity scores between individuals), which is still a weak foundation for a strong claim.

4. The simulations showed that the indices can work under ideal, controlled conditions. However, they do not prove that the indices are the best or most clinically useful tools, especially compared to established dynamic measures applied to the same real-world data.

5. The results are presented clearly, but their interpretation is overstated. The design compared DNS/DNES only to traditional static metrics. To truly validate a new dynamic metric, it must be compared against other established dynamic measures (e.g., state dwell times, transition numbers, other existing dynamic similarity measures). Without this, the claim of superiority is unsubstantiated. The authors should tone down their claims. The simulation results support the conclusion that DNS is sensitive to the four tested dynamic features and that DNES is specifically sensitive to temporal features. The real-world results suggest that the indices can detect a difference between ST and DT groups in this specific dataset. But this can only serve as a proof-of-concept based on a very small sample. The DNS and DNES metrics appear to be promising tools in a very small, specific sample. Their general effectiveness requires validation in larger, independent cohorts and against other dynamic methods.

Author Response

We would like to sincerely thank all the reviewers for their careful evaluation of our manuscript and their constructive comments. We appreciate their recognition of the novelty and methodological rigor of our work, and we are grateful for the insightful suggestions that help improve the quality of the manuscript highlighting important conceptual, methodological, and interpretative aspects. Please see below our point-to-point feedback for every comment raised by the reviewers.

Comments and Suggestions for Authors

The manuscript proposed two indices, DNS and DNES, for comparing dynamic brain networks. While the intent to move beyond static snapshot analysis is commendable and clinically relevant, the study is undermined by an underrepresented literature review that fails to establish true novelty. There are also some methodological oversights and statistical limitations. 

Answer: Thank you for the comments. Please see below our detailed point-by-point feedback, and particularly the answers to comments #1 and #2.

  1. The authors have either missed or chosen not to engage with a body of literature that directly prefigures their work. "DNS" is not a new name. The 2022 paper by He et al. (Sheng Wu Yi Xue Gong Cheng Xue Za Zhi) proposed the exact same acronym "DNS" (Dynamic Network Similarity) for the same purpose (comparing dynamic networks in stroke patients undergoing tDCS). The authors must explicitly cite this paper and clarify how their DNS formulation is mathematically or conceptually distinct. 

Answer: We thank the reviewer for pointing this out. Indeed, the 2022 paper by He et al. (published in Sheng Wu Yi Xue Gong Cheng Xue Za Zhi) was our previous publication reporting preliminaries of the same project, which contained only the DNS measure. Because that paper was not on an international scientific journal with only limited local readership, we neglected it. The present manuscript is a full report with an extensive coverage of the entire project. In particular, the current paper has included a new index DNES that was designed to intentionally isolate the temporal dynamics so that the two indices now compose a comprehensive set of tools for measuring the dynamics of evolving brain networks either structurally and temporally in combination or separately, as needed with flexibility. We have now clarified in detail the distinction of the two indices (Lines 590-595 and 651-663, marked as R2C1(for Reviwer#2 Comment #1)) and explained the current work is an extension of the previous work. The previous work is now explicitly cited as reference [40] in the paper (Lines 145 & 342 & 654).

  1. The field of dynamic network comparison is well established. The authors framed the problem as if no methods exist for comparing dynamic networks. This is incorrect. For example, the review by Mheich et al. (2020) is a dedicated summary of "brain network similarity" methods. The framework by Fu et al. (2018) and the clinical application by Fiorenzato et al. (2019) demonstrated that comparing dynamic states (e.g., dwell time, frequency, transition number) is a standard approach. While DNES is a different implementation, the core idea of quantifying temporal properties for clinical comparison is well-established. The only defensible novelty lies in the specific mathematical formulation of the indices: the particular way they combine Pearson correlation and standard deviation across the entire vector (DNS) or per-edge time series (DNES). The authors must pivot their claim of novelty from the idea of dynamic similarity to the specific performance of their unique algorithmic implementation.

Answer: We appreciate the reviewer’s valuable comment. In the revision, we have explicitly acknowledged this background and cited the relevant studies, including the review by Mheich et al. (2020) summarizing existing brain network similarity methods, the dynamic functional network connectivity (dFNC) framework by Fu et al. (2018), and its clinical application by Fiorenzato et al. (2019). These references have been added to the Discussion section to acknowledge prior work on dynamic network comparison and to clarify that the novelty of our study lies in the specific implementation of DNS and DNES (Lines 603-627).

  1. The proposed metrics have theoretical ambiguities. DNS conflates temporal and structural information into a single score. The long vector V jumbles together the evolution of each connection over time. A high DNS score could mean two networks have similar structures but are out-of-phase, or have different structures but evolve in a similar way. This makes the score difficult to interpret clinically. What does a DNS of 0.7 actually mean? Is the brain network structurally similar, temporally synchronized, or both? By comparison, state-based analyses (Fiorenzato et al., 2019) are more interpretable: "Patients dwell longer in a segregated state" provides a clear, testable hypothesis about brain function. Similarly, by calculating similarity for each edge and then averaging, DNES is highly sensitive to noise in individual connections. A few noisy or artifact-laden edges can disproportionately drag down the entire similarity score. It assumes that all connections are equally important for measuring "evolutionary similarity," which is likely false. The DNES approach appears to lack the robustness of methods that first identify major network states (like the dFNC framework) and then compare the properties of those robust, aggregate states. The claim that values of 0.5-1.0 represent "high similarity" is a bit arbitrary and not validated on multiple datasets. Conducting additional analyses to show what drives differences in DNS and DNES (e.g., which connections or time periods contribute most to low similarity) would make the indices more useful.
    Answer
    : Thank you for this constructive comment. Indeed, the measure DNS is a mixture assessment of both structure and temporal changes of the networks. This is also precisely one of the reasons that we introduce DNES which may isolate the temporal dynamics but is insensitive to structure changes. It was designed in such a way so that using them in combination would provide better insight into the dynamic networks under studying as to whether a high DNS means both structural and temporal similarity or only one of the two aspects. We have revised the manuscript to address the theoretical ambiguities (Lines 590-595 &651-663). In addition, we have removed the description of arbitrary thresholds such as “0.5–1.0 represents high similarity” to avoid misinterpretation (Lines 188-190 & 210-213). In addition, we have also discussed possible extension of applying the indices to measure networks that contain edges with unequal importance (Lines 684-699), which we plan to report in a future publication.

  1. The methods section does not justify why the specific mathematical formulation was chosen. Why use a sine wave for simulation? Why is the threshold for "high similarity" set at 0.5? The method for binarizing networks for traditional metrics is mentioned, but the critical choice of threshold (statistical significance after Bonferroni correction) may itself bias the results, as binarization is a known limitation. The real-world fMRI data analysis shows weakness of the study. It started with 25 patients and then excluded 7, leaving only 18. These were then split into tDCS1 (n=6), tDCS2 (n=5), and sham (n=7). Conducting group comparisons with such small, underpowered subgroups is a major statistical flaw. The significant p-values are suspect and likely overfit; the results are unlikely to be replicable. The use of both parametric (t-test) and non-parametric (Mann-Whitney U, Wilcoxon) tests without a clear justification for each choice raises concerns about p-hacking. The primary finding, comparing ST vs. DT groups, uses a Mann-Whitney U test on only 30 vs. 42 data points (similarity scores between individuals), which is still a weak foundation for a strong claim.

Answer: Thank you for the comment which allows us to examine our work better from other readers’ perspective. The sine wave was used because it is a straightforward approach to clearly observe the impact of phase, amplitude and translational differences on the brain networks. While other more complicated forms were available, additional nonlinear factors may be introduced to make the evaluation more complex but could defeat the effort of clarifying the source of disturbance. The revision now has explained this consideration (Lines 303-305). Please also see our answer to the previous comment, that we have removed the interpretation of 0.5 similarity threshold (Lines 188-190 & 210-213). Also, we have added to the discussion in the Limitations section noting that the relatively small sample size limits statistical power and generalizability. We also highlight the need for larger follow-up studies to confirm these findings (Line 781-787). Finally, we have toned down the claim by indicating the proposed method could be an alternative available to the research community, calling for further validations (Lines 32 &108). We will also release online the software package as a freeware welcome test and use of it (Lines 826-827).

  1. The simulations showed that the indices can work under ideal, controlled conditions. However, they do not prove that the indices are the best or most clinically useful tools, especially compared to established dynamic measures applied to the same real-world data.

Answer: Thank you for the comment. We have revised the manuscript to indicate that the 2022 publication was a subset report of this manuscript by the same group (please see our feedback to Reviewer 2 Comment #1). We have also toned down the claims regarding the superiority or clinical utility of our indices (Lines758-761).

  1. The results are presented clearly, but their interpretation is overstated. The design compared DNS/DNES only to traditional static metrics. To truly validate a new dynamic metric, it must be compared against other established dynamic measures (e.g., state dwell times, transition numbers, other existing dynamic similarity measures). Without this, the claim of superiority is unsubstantiated. The authors should tone down their claims. The simulation results support the conclusion that DNS is sensitive to the four tested dynamic features and that DNES is specifically sensitive to temporal features. The real-world results suggest that the indices can detect a difference between ST and DT groups in this specific dataset. But this can only serve as a proof-of-concept based on a very small sample. The DNS and DNES metrics appear to be promising tools in a very small, specific sample. Their general effectiveness requires validation in larger, independent cohorts and against other dynamic methods.

Answer: Thank you for your detailed and constructive comments. We have revised the manuscript to tone down the interpretation of our results, inserted into the text the statements as advised by this reviewer (Lines 597-602). In addition, we have also compared the use of the indices with other established dynamic measures, highlighting the relative strengths and limitations of our approach (Discussion section, Lines 603–627). We have also addressed the small sample size in the Limitations section, noting that the small size of the datasets limited statistical power and generalizability, and emphasizing the need for larger follow-up studies to validate these findings (Lines 781–786).

Reviewer 3 Report

Comments and Suggestions for Authors

The paper presents two novel indices, Dynamic Network Similarity (DNS) and Dynamic Network Evolution Similarity (DNES), for quantifying similarity in evolving brain networks. The authors combine simulated and real fMRI data to validate their methods, specifically using longitudinal datasets from stroke patients undergoing tDCS rehabilitation. The topic is timely and relevant, addressing the current methodological gap in dynamic brain network analysis.

Overall, the paper is technically sound, clearly written, and proposes an innovative framework for dynamic network comparison. However, several conceptual, methodological, and interpretative aspects could be clarified or strengthened.

Major Comments

  1. Conceptual clarity and positioning
    • The motivation for developing DNS and DNES is well articulated, yet the paper would benefit from a more explicit comparison with recent dynamic connectivity approaches such as time-varying graph metrics, dynamic community detection, or mutual information–based measures. It is not fully clear how DNS/DNES advance beyond existing dynamic functional connectivity (dFC) metrics in capturing temporal dependencies across individuals.
    • The relationship between DNS and DNES could be better distinguished conceptually. At present, the text gives the impression that DNES is merely a temporally focused subset of DNS, but it would help to discuss their potential complementarity and whether they could be combined into a unified similarity metric.
  2. Mathematical formulation and normalization
    • Equations (4) and (6) are central to the paper. However, the normalization range between 0 and 1 appears somewhat arbitrary and may mask meaningful variability. The authors should justify why similarity values near 0.5 are interpreted as the threshold between “low” and “high” similarity rather than providing confidence intervals or effect size–based cutoffs.
    • The decision to use Pearson correlation and standard deviation ratios as the core of both metrics is reasonable but assumes linear relationships and homoscedasticity across time. Discussion of potential non-linear extensions (e.g., Spearman, mutual information, or dynamic time warping) would strengthen the methodological contribution.
  3. Simulation design and validation
    • The simulation experiments (Experiments 1–6) are systematically designed, but the biological interpretability of the controlled parameters (Δφ, λ, α, β) is limited. It would be valuable to relate these synthetic manipulations to realistic neural phenomena, such as phase shifts in oscillatory coupling or amplitude modulations in BOLD dynamics.
    • The paper would benefit from reporting confidence intervals or standard errors alongside the reported correlations (r) and RSS values to indicate variability across the 200 or 1000 repetitions.
    • Although noise robustness is mentioned, the Gaussian noise model does not fully capture the characteristics of fMRI data. A short discussion on temporal autocorrelation and physiological noise would make the simulations more realistic.
  4. Real-world validation
    • Using stroke and tDCS rehabilitation data is an appropriate choice for testing the method in longitudinal designs. Nonetheless, the number of subjects (n=25 before exclusions) is small, especially given that six participants were removed for motion or missing data. The statistical power for between-group comparisons may thus be limited, and non-parametric bootstrapping could be used to reinforce robustness.
    • It would help to provide effect sizes and confidence intervals for the DNS and DNES differences between same-treatment (ST) and different-treatment (DT) groups. The current presentation (mainly z and p values) limits interpretation of the magnitude of group differences.
    • The binarization of networks using p<0.05 (Bonferroni corrected) is a very stringent step that may remove weak but meaningful connections. Since DNS and DNES work on weighted matrices, it would be preferable to apply them directly to continuous connectivity values rather than thresholded networks.
  5. Interpretation of results
    • The discussion occasionally overstates the generalizability of DNS/DNES (“clinicians may easily determine rehabilitation status”). These claims should be tempered; the current dataset is too limited to support direct clinical application.
    • The interpretation that DNES is “immune” to structural change (line 565-567) might be misleading. Immunity should be demonstrated mathematically rather than inferred from non-significant correlations.
    • The authors might discuss how these indices could be used to explore inter-individual variability or to identify patient subgroups with similar recovery trajectories.

Minor Comments

  • Figures could be improved for clarity: axes labels are small, and color scales for DNS vs. DNES should be consistent across panels.
  • The introduction repeats background on static network analysis extensively; condensing it would improve focus.
  • Several typographical errors appear (“therefor,” “praradigm”) and should be corrected.
  • It would help to include pseudo-code or a simple algorithmic description of DNS and DNES for reproducibility.
  • The authors should deposit a small sample dataset and code (e.g., MATLAB or Python scripts) for transparency.

Author Response

We would like to sincerely thank all the reviewers for their careful evaluation of our manuscript and their constructive comments. We appreciate their recognition of the novelty and methodological rigor of our work, and we are grateful for the insightful suggestions that help improve the quality of the manuscript highlighting important conceptual, methodological, and interpretative aspects. Please see below our point-to-point feedback for every comment raised by the reviewers.

Comments and Suggestions for Authors

The paper presents two novel indices, Dynamic Network Similarity (DNS) and Dynamic Network Evolution Similarity (DNES), for quantifying similarity in evolving brain networks. The authors combine simulated and real fMRI data to validate their methods, specifically using longitudinal datasets from stroke patients undergoing tDCS rehabilitation. The topic is timely and relevant, addressing the current methodological gap in dynamic brain network analysis.

Overall, the paper is technically sound, clearly written, and proposes an innovative framework for dynamic network comparison. However, several conceptual, methodological, and interpretative aspects could be clarified or strengthened.

Answer: Thank you for the comments. Please see below our detailed point-by-point feedback.

Major Comments

  1. Conceptual clarity and positioning
    • 1. The motivation for developing DNS and DNES is well articulated, yet the paper would benefit from a more explicit comparison with recent dynamic connectivity approaches such as time-varying graph metrics, dynamic community detection, or mutual information–based measures. It is not fully clear how DNS/DNES advance beyond existing dynamic functional connectivity (dFC) metrics in capturing temporal dependencies across individuals.

Answer: Thank you for your comment. Previous methods such as dFC use a sliding window with the same time series of fMRI data and measures the state of the brain state within that time series. In contrast, our proposed DNS & DNES are for measuring brain network dynamics in longitudinal data across multiple time points and different studies. We now have elaborated in the text comparisons with recent dynamic connectivity methods, emphasizing that DNS and DNES provide complementary information by capturing whole-network and edge-level temporal similarity across individuals at multiple time points (Lines 603-627, marked as R3C1.1(for Reviwer#3 Comment #1.1)).

  • 2. The relationship between DNS and DNES could be better distinguished conceptually. At present, the text gives the impression that DNES is merely a temporally focused subset of DNS, but it would help to discuss their potential complementarity and whether they could be combined into a unified similarity metric.

Answer: Thank you for your insightful comment. We have revised the manuscript to clarify the conceptual distinction and complementarity between DNS and DNES (Lines 590-594 and 659–667 in the updated text). Specifically, DNS may be seen as quantifying combined temporal and structural similarity at the whole-network level, whereas DNES captures temporal similarity at the level of individual connections.  Please also refer to our answer to Reviewer 1 Comment #3.

  1. Mathematical formulation and normalization
    • 1. Equations (4) and (6) are central to the paper. However, the normalization range between 0 and 1 appears somewhat arbitrary and may mask meaningful variability. The authors should justify why similarity values near 0.5 are interpreted as the threshold between “low” and “high” similarity rather than providing confidence intervals or effect size–based cutoffs.

Answer: Thank you for your comment. The normalization between 0-1 is for comparability across different studies, however, it is not supposed for categorical interpretation. We therefore have removed the descriptions regarding the 0.5 threshold. Please see Lines 188-190, 210-213, 623-625; and also our answers to Reviewer 2 Comments #3&#4.

  • 2. The decision to use Pearson correlation and standard deviation ratios as the core of both metrics is reasonable but assumes linear relationships and homoscedasticity across time. Discussion of potential non-linear extensions (e.g., Spearman, mutual information, or dynamic time warping) would strengthen the methodological contribution.

Answer: Thank you for your comment. We have noted in the limitations section that DNS and DNES rely on Pearson correlation, which assumes linearity and homoscedasticity, and that future work should also explore non-linear alternatives to incorporate metrics such as Spearman correlation, mutual information, or dynamic time warping (Lines 787-792).

  1. Simulation design and validation
    • 1. The simulation experiments (Experiments 1–6) are systematically designed, but the biological interpretability of the controlled parameters (Δφ, λ, α, β) is limited. It would be valuable to relate these synthetic manipulations to realistic neural phenomena, such as phase shifts in oscillatory coupling or amplitude modulations in BOLD dynamics.

Answer: Thank you for your comment. We now have added explanations that the simulation parameters (Δφ, λ, α, β) were chosen to reflect physiologically plausible neural phenomena—phase evolution, amplitude modulation, noise levels, and connectivity strength—and were empirically validated through pilot simulations to ensure sensitivity and statistical stability (Lines 328-342).

  • 2. The paper would benefit from reporting confidence intervals or standard errors alongside the reported correlations and RSS values to indicate variability across the 200 or 1000 repetitions.

Answer: Thank you for your comment. We have added confidence intervals to the reported results to reflect variability across simulation repetitions (Lines 514 & 524).

  • 3. Although noise robustness is mentioned, the Gaussian noise model does not fully capture the characteristics of fMRI data. A short discussion on temporal autocorrelation and physiological noise would make the simulations more realistic.

Answer: Thank you for your comment. We explained that the noise in MRI data typically follows a Rician distribution. However, it reduces to Gaussian when SNR is higher than 2.0 or ~3.0 dB, and included a reference [41](Gudbjartsson H, Patz S. The Rician distribution of noisy MRI data. Magn Reson Med 1995;34(6):910–4).  Please see Lines 324-326. Nevertheless, we understand that brain networks derived from the fMRI data may not inherit the noise and exactly follow the same distribution, therefore, we have added to the limitation section a note (the fourth) to address this concern (Lines792-796).

  1. Real-world validation
    • 1. Using stroke and tDCS rehabilitation data is an appropriate choice for testing the method in longitudinal designs. Nonetheless, the number of subjects (n=25 before exclusions) is small, especially given that six participants were removed for motion or missing data. The statistical power for between-group comparisons may thus be limited, and non-parametric bootstrapping could be used to reinforce robustness.

Answer: Thank you for this insightful comment. We fully acknowledge the reviewer’s concern regarding the limited sample size, especially after data exclusions, which may reduce the statistical power of between-group comparisons. In the revised manuscript, we have expanded the Discussion to explicitly address this limitation and its impact on generalizability. We have also noted potential remedies for future work, including increasing the sample size and incorporating non-parametric bootstrapping approaches, as suggested. Please see Lines 781–787 in the revised version.

  • 2. It would help to provide effect sizes and confidence intervals for the DNS and DNES differences between same-treatment (ST) and different-treatment (DT) groups. The current presentation (mainly z and p values) limits interpretation of the magnitude of group differences.

Answer: Thank you for your comment. We now report effect sizes (Cohen’s d) and 95% confidence intervals alongside DNS and DNES group comparisons (Lines 498-500, and 585-586).

  • 3. The binarization of networks using p<0.05 (Bonferroni corrected) is a very stringent step that may remove weak but meaningful connections. Since DNS and DNES work on weighted matrices, it would be preferable to apply them directly to continuous connectivity values rather than thresholded networks.

Answer: Thank you for this insightful suggestion. We planned this paper as the first one that reports the basic framework of both the 2 indices as a comprehensive set of tools for measuring brain network dynamics, and plan to report their further extensions concerning weighted and directional connectivity in a follow-up paper, as a lot of technical details will need to be added for this extension and the page limit of the current paper will not allow the needed space. However, to meet the reviewer’s expectation, we now have added a paragraph elaborating and discussing this consideration. Please see Lines 797-801.

  1. Interpretation of results
    • 1. The discussion occasionally overstates the generalizability of DNS/DNES (“clinicians may easily determine rehabilitation status”). These claims should be tempered; the current dataset is too limited to support direct clinical application.

Answer: Thank you for pointing this out. This statement has been removed. Moreover, we have revised the manuscript to tone down the claims regarding clinical generalizability (first paragraph of Discussion Lines 597-602 and Lines 727-740). We also note in the limitations section that while DNS and DNES showed promise, the current dataset was small and the findings were preliminary. Therefore, these metrics yet need to be further validated using larger and more diversified data. (Lines758-761).

  • 2. The interpretation that DNES is “immune” to structural change (line 565-567) might be misleading. Immunity should be demonstrated mathematically rather than inferred from non-significant correlations.

Answer: Thank you for your comment. We have removed the claim that DNES is “immune” to structural change and now describe it as primarily sensitive to temporal similarity, with relative insensitivity to structural variations (Lines 642-644)

  • 3. The authors might discuss how these indices could be used to explore inter-individual variability or to identify patient subgroups with similar recovery trajectories.

Answer: We thank the reviewer for this insightful suggestion. Yes, DNS and DNES have framed a way to explore inter-individual variability and to identify patient subgroups with similar recovery trajectories. In the Discussion, we have added descriptions on how DNS/DNES may be used to infer inter-individual variability or to identify patient subgroups with similar recovery trajectories (Lines 727-740).

Minor Comments

  1. Figures could be improved for clarity: axes labels are small, and color scales for DNS vs. DNES should be consistent across panels.

Answer: Thank you for the helpful suggestion.  Following the reviewer’s opinions, we have revisited all the figures and updated the labels and color agendas and updated them with much better visibility. Please see updated figures 1, 2, 3, 4, 5.

  1. The introduction repeats background on static network analysis extensively; condensing it would improve focus.

Answer: Thanks for pointing out this. We now have shortened the background description (Lines 65-71)

  1. Several typographical errors appear (“therefor,” “praradigm”) and should be corrected.

Answer: We apologize for these typos. Fixed. Thank you.

  1. It would help to include pseudo-code or a simple algorithmic description of DNS and DNES for reproducibility.

Answer: Thank you for the valuable suggestion. While pseudo-code was not added to the manuscript, the full implementation of DNS and DNES will be made publicly available as supplementary material to facilitate reproducibility. Please see the Data Availability Statement (Lines 826-827).

  1. The authors should deposit a small sample dataset and code (e.g., MATLAB or Python scripts) for transparency.

Answer: Thank you for the valuable suggestion. The full implementation of DNS and DNES will be made publicly available as supplementary material to facilitate reproducibility. Please see the Data Availability Statement (Lines 826-827).

Round 2

Reviewer 1 Report

Comments and Suggestions for Authors

The manuscript has been sufficiently improved to warrant publication in Bioengineering. I approve revised version for publication in Bioengineering. 

Reviewer 2 Report

Comments and Suggestions for Authors

Thank you for your detailed responses and revision. I recommend acceptance.